# Temporal trends in incidence of atrial fibrillation in primary care records: a population-based cohort study

Sílvia C Mendonça [ID] , Catherine L Saunders, Jenny Lund, Jonathan Mant, Duncan Edwards

► Prepublication history additional material for this paper is available online. To view these files, please visit the journal online (http://dx.doi.org/10.1136/bmjopen-2020-042518).

Primary Care Unit, Department of Public Health and Primary Care, University of Cambridge, Cambridge, UK

**Correspondence to**
Sílvia C Mendonça;
sm2061@medschl.cam.ac.uk

## ABSTRACT

**Objectives** Atrial fibrillation (AF) is a heart condition associated with a fivefold increased risk of stroke. The condition can be detected in primary care and treatment can greatly reduce the risk of stroke. In recent years, a number of policy initiatives have tried to improve diagnosis and treatment of AF, including local National Health Service schemes and the Quality and Outcomes Framework. We aimed to examine trends in the incidence of recorded AF in primary care records from English practices between 2004 and 2018.

**Design** Longitudinal cohort study.

**Setting** English primary care electronic health records linked to Index of Multiple Deprivation data.

**Participants** Cohort of 3.5 million patients over 40 years old registered in general practices in England, contributing 22 million person-years of observation between 2004 and 2018.

**Primary and secondary outcome measures** Incident AF was identified through newly recorded AF codes in the patients' records. Yearly incidence rates were stratified by gender, age group and a measure of deprivation.

**Results** Incidence rates were stable before 2010 and then rose and peaked in 2015 at 5.07 (95% CI 4.94 to 5.20) cases per 1000 person-years. Incidence was higher in males (4.95 (95% CI 4.91 to 4.99) cases per 1000 person-years vs 4.12 (95% CI 4.08 to 4.16) in females) and rises markedly with age (0.58 (95% CI 0.56 to 0.59) cases per 1000 person-years in 40–54 years old vs 21.7 (95% CI 21.4 to 22.0) cases in over 85s). The increase in incidence over time was observed mainly in people over the age of 75, particularly men. There was no evidence that temporal trends in incidence were associated with deprivation.

**Conclusions** Changes in clinical practice and policy initiatives since 2004 have been associated with increased rates of diagnosis of AF up until 2015, but rates declined from 2015 to 2018.

## Strengths and limitations of this study

► Our population-based sample covered 3.5 million patients at risk over a period of 15 years.
► The data from the Clinical Practice Research Datalink used in this study have been shown to be representative of the UK population and to be of high quality for the recording of diagnoses, particularly for chronic conditions.
► Atrial fibrillation (AF) has been part of the Quality and Outcomes Framework since 2007 and is likely to be well recorded in primary care records.
► A limitation is that these reported incidence rates exclude patients with undiagnosed AF, or that do not have their diagnosis recorded in their primary care records.

## INTRODUCTION

Atrial fibrillation (AF) is a cardiac arrhythmia that is associated with a fivefold increase in risk of stroke.[1] The risk of stroke associated with AF can be greatly reduced by use of anticoagulants. AF is associated with 1 in 5 stroke cases[2] and is considered to be underdiagnosed[3] and undertreated.[4] Consequently, AF has been the target of a number of policy

initiatives in recent years aimed at both treatment and diagnosis. In the UK, it has been part of the Quality and Outcomes Framework (QOF) since 2006,[5] the pay-for-performance scheme that incentivises the recording and appropriate management of chronic conditions. Since 2008, there have been several local National Health Service (NHS) schemes to incentivise detection and management of AF including the Guidance on Risk Assessment and Stroke Prevention for Atrial Fibrillation electronic tool[6] and opportunistic screening by manual pulse checks.[5] It is the subject of a national programme coordinated by the Academic Health Sciences Network.[2] The need for better detection is highlighted in the NHS Long Term Plan (2019).[7] There is evidence from both UK and international studies of a rise in the age-specific incidence of AF,[8 9] including two using UK electronic primary care records.[10 11] However, these have not used data beyond 2011. Given that policy initiatives continue to focus on AF, our aim was to describe temporal trends in recorded AF in a nationally representative sample of primary care records up until 2018.

## METHODS

### Data source

We used data from the Clinical Practice Research Datalink (CPRD) GOLD, a database of electronic primary care records in the UK, based on patients attending general practices which use the Vision computer system. CPRD has been shown to be nationally representative of the UK population[12] and data quality is monitored at the practice and patient level. In 2013, it contained the medical records of over 11 million patients, with 4.4 active (currently registered) patients corresponding to about 7% of the UK population.[12] During the study period (2004–2018) the number of UK practices contributing data to CPRD varied from 680 to 367 (online supplemental appendix 1) due to a reduction of practices using the Vision computer system. The data collected from CPRD include all clinical codes for medical diagnoses entered at consultations, referrals, tests and all prescriptions issued at the practice.

### Study cohort and disease definition

The study population included patients from English practices aged over 40 years between 1 January 2004 and 31 December 2018. Recorded AF was identified using a list of eight Read codes (online supplemental appendix 2). This list was developed as part of a project on the epidemiology of multimorbidity[13 14] and was based originally on the QOF business rules. We used the October 2019 data release of CPRD GOLD.

### Data analysis

AF incidence was estimated as the number of new AF cases during each year, divided by the eligible follow-up (person-time at risk). Eligible follow-up for all patients (with and without AF) was calculated using the CPRD denominator files, which contain basic information about each patient and practice. Start of follow-up was calculated as the latest of 1 year after the patient first registered at the practice, the practice up to standard date (an internal measure of CPRD data quality based on consistency of data provided to CPRD by the practice[12]) and 1 January 2004 (study start). The end of follow-up was the earliest of the last date the patient was registered at the practice, date of death, the last date the practice contributed data to CPRD, or 31 December 2018. We restricted follow-up to start 1 year after registration at the practice to avoid capturing historical diagnosis.

People with a new AF medical code recorded for the first time during the eligible follow-up period were identified as incident cases and counted towards the numerator. Those people with any AF code identified before the start of eligible follow-up were considered prevalent cases and were excluded. Cases registered after the end of eligible follow-up were also excluded from the numerator, except for a few cases that occurred within 3 months of death or transfer out date. We decided to include these as they probably relate to events that occurred within eligible follow-up but that were registered later. AF cases that occurred within or after eligible follow-up contribute time at risk to the denominator until their date of diagnosis or end of follow-up, whichever is first.

Incidence rates are calculated as the number of new cases of AF in our eligible population and study period (numerator) divided by the total person-time at risk (denominator). The denominator was the sum of the person-time at risk for all eligible patients (cases and non-cases).

The patients' sex and year of birth were available from CPRD. We assumed the patients' date of birth to be the midyear (ie, 30 June) of year of birth. Patients' age was grouped into five categories: 40–54, 55–64, 65–74, 75–84 and ≥85 years old.

Stratified incidence rates by calendar year and sociodemographic characteristics were calculated through the use of a Lexis expansion.[15] Each patient-time at risk was split into multiple time periods according to their age group and calendar years. Time at risk is then added up within each stratum, a combination of calendar year by age group by gender (eg, women aged 65–74 in 2016). The incidence rate corresponds to the number of AF cases divided by the total person-time at risk in that stratum. P values for the crude effects were derived from unadjusted Poisson models; p values for the effect of age and gender were calculated using a gender-stratified model and a model with the two-way interaction.

In order to take into account possible changes over time in the age composition of the sample, we have additionally estimated modelled incidence rates. We used a Poisson regression approach using the number of events in each stratum as the main outcome and including the logarithm of follow-up time as an offset. The model for all patients included the main effect variables year, age group and sex, and their two-way and three-way interactions. Year and sex were always treated as categorical variables but age group was treated as a continuous variable for the interaction terms only in order to maximise power. Each model was used to predict the incidence rate for each year adjusting for the case mix characteristics of the whole sample (known as marginal standardisation).[16]

Incidence by deprivation was calculated using a CPRD patient denominator file stratified by the patient-level 2010 English Index of Multiple Deprivation (IMD) quintiles and the linked patient-level IMD quintiles for the patients with newly registered AF. For this data linkage, data are only available until the end of 2016. To model the effect of deprivation we used the same model as the main analysis but adding in a categorical main effect for deprivation quintile and an interaction between year and deprivation quintile (treated as a linear term).

Data manipulation and analyses were performed in Stata V.15.1 (StataCorp). We report 95% CIs calculated as exact Poisson CIs.

### Patient and public involvement

Patients and the public were not involved in this study.

**Table 1** Characteristics of patients with newly diagnosed AF in the period 2004–2018

| | n (%) |
|---|---|
| All patients | 99 836 |
| Gender | |
| Men | 52 823 (52.9) |
| Women | 47 013 (47.1) |
| Age, mean (SD) | 74.6 (11.2) |
| Age group | |
| 40–54 | 5597 (5.6) |
| 55–64 | 12 598 (12.6) |
| 65–74 | 26 308 (26.4) |
| 75–84 | 36 215 (36.3) |
| 85+ | 19 118 (19.1) |

AF, atrial fibrillation.

## RESULTS

In a study population of nearly 3.5 million patients contributing a total of 22 million person-years of observation, there were 99 836 newly registered AF cases (see patient selection flow chart in online supplemental appendix 3) with a mean age at diagnosis of 74.6 years (see table 1). Over 55% of new diagnoses were in people over the age of 75. The overall incidence is 4.52 cases per 1000 patient-years in people aged over 40 years, and 11.6 per 1000 patient-years in people aged 65 years and older. The incidence rises by age, up to 21.7 per 1000 in people aged 85 or over, and is higher in men than women, 4.95 vs 4.12 per 1000 person-years (table 2). There was evidence of significant variation associated with the effects of age and gender (p<0.001 for both, table 2). We also noted that incidence increases with age differently for men and women (p<0.001, table 2 and figure 1).

Incidence of AF was relatively stable between 2004 and 2010, between 4.2 and 4.5 cases per 1000 person-years, then increased up until 2015, when incidence peaked at 5.1 cases per 1000 person-years, before falling back to 4.6 cases per 1000 person-years by 2018 (see figure 1 and online supplemental appendix 4). There was strong evidence of variation between years (p<0.001). This post-2015 decline occurred primarily in women. The age-specific trends show that the increase between 2010 and 2015 occurred largely in people aged 75 and over, and more so in men (figure 2 and online supplemental appendix 5).

Incidence of AF was higher in the most deprived quintile of IMD than in the least deprived quintile (p=0.002 for the effect of deprivation; online supplemental appendices 6 and 7). There was no evidence of differences in temporal trends in AF by levels of socioeconomic deprivation (p=0.182).

**Table 2** Observed incidence rates (per 1000 person-years) for recorded AF during the period 2004–2018, stratified by age and gender

| | Incidence rate | 95% CI | P value* |
|---|---|---|---|
| All patients | 4.52 | 4.49 to 4.55 | |
| Gender | | | <0.001 |
| Female | 4.12 | 4.08 to 4.16 | |
| Male | 4.95 | 4.91 to 4.99 | |
| Age group | | | <0.001 |
| 40–54 | 0.58 | 0.56 to 0.59 | |
| 55–64 | 2.36 | 2.32 to 2.40 | |
| 65–74 | 6.84 | 6.76 to 6.93 | |
| 75–84 | 15.5 | 15.3 to 15.7 | |
| 85+ | 21.7 | 21.4 to 22.0 | |
| Gender and age group | | | <0.001 |
| Women | | | <0.001 |
| 40–54 | 0.32 | 0.30 to 0.34 | |
| 55–64 | 1.52 | 1.48 to 1.57 | |
| 65–74 | 5.24 | 5.14 to 5.34 | |
| 75–84 | 13.7 | 13.5 to 13.9 | |
| 85+ | 20.3 | 20.0 to 20.7 | |
| Men | | | <0.001 |
| 40–54 | 0.83 | 0.80 to 0.85 | |
| 55–64 | 3.20 | 3.14 to 3.27 | |
| 65–74 | 8.61 | 8.48 to 8.75 | |
| 75–84 | 18.0 | 17.7 to 18.3 | |
| 85+ | 24.8 | 24.2 to 25.3 | |

*The p values presented in this table come from unadjusted tests of whether there is variation by age and sex (overall, top two sets of rows) and variation by age, stratified by sex (below, bottom two sets of rows). The interaction p value, however, has a slightly different interpretation and presents a measure of the strength of evidence of whether the effect of age is different for men and women (middle row).
AF, atrial fibrillation.

## DISCUSSION
### Summary

We observed stable AF incidence from 2004 to 2010, increased incidence from 2010 to 2015 and then falling incidence from 2016 to 2018. The rise in incidence was primarily in people aged 75 and over and most marked in men. The decline from 2016 to 2018 reflected falling incidence in women rather than men.

### Strengths and limitations

A strength of this study was the use of a large sample that is representative of the English population, permitting a detailed description of current practice. It is important to note that the focus of this study was *recorded* (or clinically detected) AF, and thus patients with undiagnosed, or who are diagnosed but not recorded will not be included.

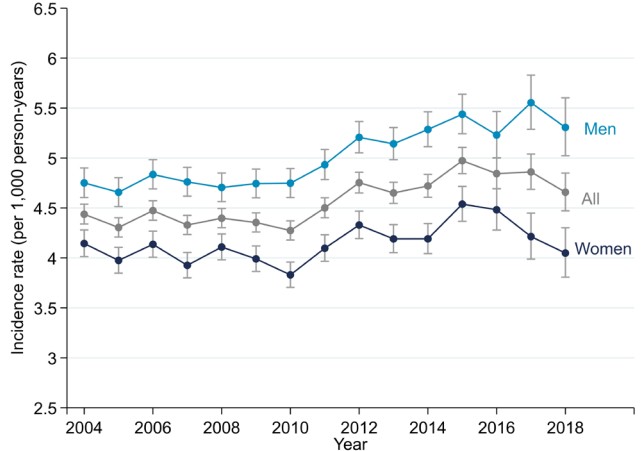

**Figure 1** Modelled incidence of recorded atrial fibrillation over time.

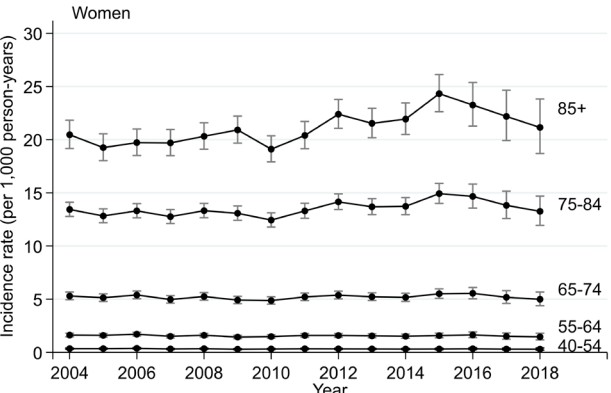

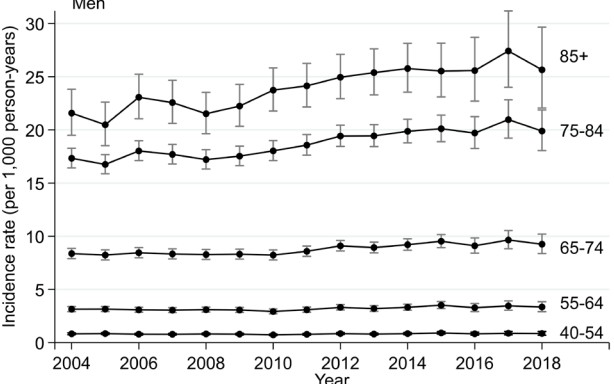

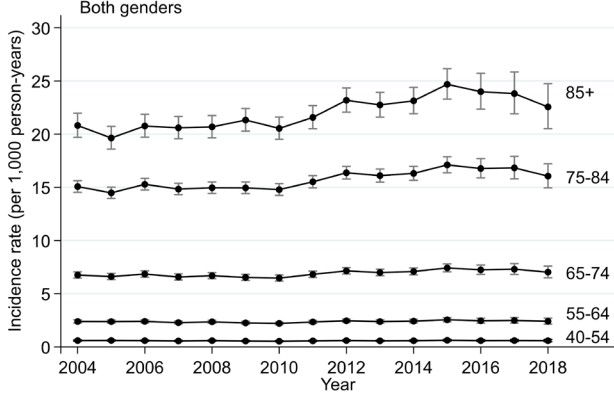

**Figure 2** Modelled incidence of recorded atrial fibrillation over time stratified by age group and gender.

However, diagnosis and inclusion in population registers is essential to ensure adequate care and stroke prevention, and the focus of this study was to describe the incidence of clinically detected AF. Our modelled incidence rates remove the effect that possible changes in the population case mix over time could have on observed rates. However, changes in the number of practices using the Vision computer system in recent years could be contributing to the levelling off in incidence rates observed after 2015.[17]

## Comparison with existing literature

Incidence rates for AF can vary considerably between different studies, most likely due to different source populations, case definitions and study periods. The rates reported in this study are similar to those described in other European cohorts from Iceland[18] and the Netherlands,[19] but lower than one from Germany.[20] In general, incidence rates reported from American cohorts[21–23] are higher than those from Europe, including our own study.

In the UK population, we found two studies that looked at incidence of AF over time.[10 11] Scowcroft and Cowie calculated incidence rates between 2000 and 2012 while Lane *et al* report incidence rates between 1998 and 2010. Both studies use data from CPRD and also report increases in incidence (Scowcroft and Cowie from 2009 onwards and Lane *et al* between the periods 1998–2001 and 2002–2006). We found higher incidence rates than those reported by Lane *et al* and Scowcroft and Cowie. Whereas we restricted our analysis to people aged 40 years and over, they both reported incidence in people aged 18 years and over. However, even when looking at comparable age groups, our rates were higher. For example, Lane *et al* reported an incidence rate of 3.26 per 1000 people aged 65–74 between 2007 and 2010, whereas the observed incidence rate in our study in this age group was 6.84 per 1000. This is likely to reflect analytical differences in how the denominator population was defined. In our study, we derived both the numerator and

the denominator by first defining the eligible follow-up period for all patients and then identifying which AF cases fell in this eligibility period. In the other two UK studies, these figures appear to have been derived separately, with stricter criteria applied to the cases (eg, Scowcroft and Cowie impose a minimum of 1 year both before and after diagnosis) than to the denominator, resulting in lower rates, though the observed temporal trends will not have been affected. We believe our estimates are truer reflections of the incidence of AF. A recent meta-analysis of AF screening studies in a general population estimated a detection rate of 1.44% in patients aged 65 and over.[24] This is slightly higher than our estimate of 1.16 cases

**Table 3** Key changes in policy and practice during the study period, 2004–2018

| Policy or evidence change | Year |
| --- | --- |
| Requirement for practice AF registers incorporated within QOF.[31] | 2007 |
| First major NOAC RCT demonstrates effectiveness.[32] | 2009 |
| Widespread introduction of local incentive schemes (LES) to diagnose and/or treat AF.[33] | 2011 |
| Expansion of AF within QOF from three to four indicators.[31] | 2013 |
| NICE guidance suggests a larger proportion of patients with AF are eligible for anticoagulation.[34] | 2014 |

AF, atrial fibrillation; LES, local enhanced service; NICE, National Institute for Health and Care Excellence; NOAC, novel oral anticoagulants; QOF, Quality and Outcomes Framework; RCT, randomised controlled trial.

per 100 person-years in the comparable age group (see online supplemental appendix 8), which is what would be expected given that these were screening studies, whereas ours would have identified only those picked up from active case finding.[3] Further corroborative evidence that our incidence rates are more likely to be correct comes from the Framingham study, which reported incidence rates of 13.4 per 1000 in men over the age of 50, and 8.6 per 1000 in women. The higher rates in Framingham study, despite the slightly younger age group, reflect the multiple sources of ascertainment that were used to identify AF. Detection rates from routine practice will be lower than those.

Ours is the first study to demonstrate a decline in the incidence of recorded AF between 2015 and 2018 in the UK, after taking account of changes over time in age composition of the population. Given that epidemiological studies such as Framingham have reported increases rather than decreases in AF incidence,[9] this is likely to reflect a change in ascertainment in recent years.

### Implications for research, policy and practice

Our results suggest that initiatives in policy and clinical practice since 2004 have been associated with increased rates of diagnosis of AF (table 3). Although the increase in incidence seen in over 85 year-olds may be explained in part by the increased mean age of this group over time, it is also plausible that initiatives to encourage pulse palpation in over 65s,[5] or improved perceived tolerability of newer anticoagulants[25] have influenced clinicians to increase efforts to diagnose older patients. It is also notable that increases in AF diagnosis do not appear to have come at the expense of the widening social inequality that has been of particular concern in conditions such as cancer.[26] The small decline in AF incidence since 2015 may reflect reducing policy emphasis or a decline in the primary care workforce available to detect AF.[27] The decline in incidence may also reflect that 'catch up'

diagnoses were made in the years prior to 2015, boosting apparent incidence during that period. Given the emergence of new technologies which allow self-diagnosis of AF, it is likely that the incidence of diagnosis of AF will rise again in the future.[28] Indeed, data from studies that have used implantable loop recorders to detect AF suggest that there may be substantial proportions of people with short episodes of undiagnosed AF.[29]

It is important to continue to monitor trends in AF incidence to assess impact of new technologies and initiatives. Our finding that incidence of AF in the under 65 age group remained low and did not change during the study period suggests first that there has been no substantive shift in the clinical presentation of this group to primary care that has led to changes in AF incidence in this population. Policy initiatives such as incentives for opportunistic screening have not focused on younger people with AF; our study would tend to support this approach as the numbers of people identified would be likely to be very low.

Our research has also highlighted the methodological point that researchers should ensure they use the correct denominators for studies of incidence in primary care database analyses as small changes in approaches can lead to substantial errors in published rates, with potentially large implications for policy decisions based on these estimates. We would recommend that policymakers base future modelling decisions and analyses on the rates presented in this, rather than previous studies.

### CONCLUSION

The incidence of diagnosed AF increased in the UK between 2004 and 2015, with the greatest increase among patients aged over 85, but with a small decline between 2015 and 2018. Trials of AF screening that are powered to detect impact on stroke rates will provide evidence on the extent to which efforts should be made to reverse this decline.[30]

**Correction notice** This article has been corrected since it first published. The provenance and peer review statement has been included.

**Acknowledgements** This study is based in part on data from the Clinical Practice Research Datalink obtained under licence from the UK Medicines and Healthcare products Regulatory Agency. The data are provided by patients and collected by the NHS as part of their care and support. The interpretation and conclusions contained in this study are those of the authors alone.

**Contributors** DE and JM had the original idea for the study. DE, SCM, CLS, JL and JM designed the study. SCM undertook data extraction and performed the analysis. SCM and DE wrote the first draft of the paper, which was revised in collaboration with all authors. DE acts as the guarantor of the study.

**Funding** This report is an independent research funded by the National Institute for Health Research (NIHR School for Primary Care Research, reference FR11/290).

**Disclaimer** The views expressed in this publication are those of the authors and not necessarily those of the NHS, the National Institute for Health Research or the Department of Health.

**Competing interests** All authors report grants from the NIHR School of Primary Care Research, during the conduct of the study. JL reports grants from the Wellcome Trust and the National Institute for Health Research, outside the

submitted work. JM is on an advisory board for BMS/Pfizer on an AF screening trial, and is chief investigator of an NIHR-funded programme grant on screening for atrial fibrillation. DE is a coinvestigator on this same NIHR programme.

**Patient consent for publication** Not required.

**Ethics approval** This study was approved by the Independent Scientific Advisory Board for Medicines and Healthcare products Regulatory Agency research (protocol reference number 17_079R). No further ethics approval is required.

**Provenance and peer review** Not commissioned; externally peer reviewed.

**Data availability statement** The study uses data from the Clinical Practice Research Datalink (CPRD). CPRD does not allow the sharing of patient-level data. The data specification for the CPRD data set is available at: https://www.cprd.com/sites/default/files/CPRD_GOLD_Full_Data_Specification_v2.0_0.pdf. Additional information on the IMD patient-level linkage data is available at: https://cprd.com/sites/default/files/Documentation_SmallAreaData_Patient_set18_v2.7.pdf. The code list used in this study is part of a set of lists made available at www.phpc.cam.ac.uk/pcu/cprd_cam/codelists/.

**Open access** This is an open access article distributed in accordance with the Creative Commons Attribution 4.0 Unported (CC BY 4.0) license, which permits others to copy, redistribute, remix, transform and build upon this work for any purpose, provided the original work is properly cited, a link to the licence is given, and indication of whether changes were made. See: https://creativecommons.org/licenses/by/4.0/.

**ORCID iD**
Sílvia C Mendonça http://orcid.org/0000-0001-5504-4906

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
