## [Reviewer comments · BMJ Open]

ARTICLE DETAILS

TITLE (PROVISIONAL)	Temporal trends in incidence of atrial fibrillation in primary care records: a population-based cohort study
AUTHORS	Mendonca, Silvia; Saunders, Catherine; Lund, Jenny; Mant, Jonathan; Edwards, Duncan

VERSION 1 – REVIEW

REVIEWER	Gurukripa N. Kowlgi, MBBS Mayo Clinic, Rochester, MN, USA
REVIEW RETURNED	18-Aug-2020

GENERAL COMMENTS	Medonca et al. have done a commendable job in their study titled “Temporal trends in incidence of atrial fibrillation in primary care records”. The authors have studied a cohort of 3.5 million patients over 40 years of age between 2004 and 2018 utilizing the Clinical Practice Research Datalink, and report the detailed trends in atrial fibrillation (AF) incidence. They report that the incidence of AF remained stable from 2004-10, then increased up until 2015, and declined from 2015-2018. The authors hypothesize that changes in clinical practice and policy initiatives have led to the increased rates of diagnosis from 2010-15. Overall this is a well-conducted study with meticulous analysis of the data. The authors describe in detail how the incidence calculations were made in the data analysis section. The difference in AF incidence in the present study compared to prior ones is quite significant, but the authors diligently explain the reasons for this. Points for revision: MAJOR: 1) In the Results section, the statements of incidence comparison need to be substantiated with p-values. These are missing in both the text and table 2.2) The decline between 2015-18 seems to be more in the older age group 75-84 and 85+ than the younger ones. Did the authors note a significant change in the denominators of these groups between 2015-18? A differentiation should be made between drop in incidence due to a drop in gross number of new cases, versus increase in the denominator, or both.3) In Table 1, the baseline characteristics for age and gender are provided. Given the extent of the available database, would it not be possible to include the frequencies of comorbidities associated with AF such as hypertension, congestive heart failure, stroke among others? This information could be useful in determining the reasons for the observed trends in AF incidence. MINOR: 1) While the hypothesis of “boosting” incidence before 2015 can be entertained, the data provided is not confirming that this is the sole
--

	reason for the decline in AF incidence in more recent years. This is the first study to record a decline in AF incidence from 2015-18. While this is very interesting, the discussion should be expanded to include other potential reasons. In addition to health policy changes, the authors could comment on other factors such as improvement in technology of cardiac implanted device detected AF. 2) In the current era of wearable-device detected AF, this decline in AF incidence will likely not sustain in the future. The authors should mention a line on this in the implications for research section.
--	--

REVIEWER	Eitaro Kodani Nippon Medical School Tama-Nagayama Hospital, Tokyo, Japan
REVIEW RETURNED	16-Sep-2020

GENERAL COMMENTS	This manuscript by Mendonça et al. focused on the incidence atrial fibrillation (AF) in primary care. Authors investigated that the incidence of AF using a large cohort of 3.5 million patients over 40 years old registered in general practices in England and demonstrated its trends between 2004 and 2018 by sex and age classes. As authors mentioned, the prevalence and incidence of AF have been increasing along with increasing aged population, and recently become a social problem. Therefore, the concept of this study is valuable and results seem reasonable. Although this manuscript seems written well, authors may want to consider several issues as below. Major comments; 1) How did you deal with the difference in observation periods among the year? For instance, was it for 14 years in the 2004's data? Whereas, how long was it 2018's data? 2) In Figure 1, if authors wanted to compare the incidence of AF between the period of 2004-2010 and that of 2010-2018, show statistical difference. 3) In Appendix 6, show statistical difference. Minor comments; 1) In abstract, abbreviations of NHS should be spelt out even in abstract. In line 2, although authors described that "The condition is easy to detect in primary care", I feel out of tune this sentence, because it seems difficult to detect paroxysmal AF in primary care. Authors may want to modify this sentence. 2) In main text, abbreviations of NHS, GRASP-AF, GOLD, and QOF should be spelt out at the first time of use. 3) In Box 1, abbreviations of QOF, NOAC, RCT, and NICE should be explained in footnote. 4) In Appendix 4 and 5, please explain shortly what "Modelled rate" is in footnote. 5) In Appendix 7, abbreviations of IMD should be spelt out.
---

VERSION 1 – AUTHOR RESPONSE

Reviewer: 1

Reviewer Name: Gurukripa N. Kowlgi, MBBS Institution and Country: Mayo Clinic, Rochester, MN, USA Please state any competing interests or state 'None declared': None declared

Mendonca et al. have done a commendable job in their study titled “Temporal trends in incidence of atrial fibrillation in primary care records”. The authors have studied a cohort of 3.5 million patients over 40 years of age between 2004 and 2018 utilizing the Clinical Practice Research Datalink, and report the detailed trends in atrial fibrillation (AF) incidence. They report that the incidence of AF remained stable from 2004-10, then increased up until 2015, and declined from 2015-2018. The authors hypothesize that changes in clinical practice and policy initiatives have led to the increased rates of diagnosis from 2010-15. Overall this is a well-conducted study with meticulous analysis of the data. The authors describe in detail how the incidence calculations were made in the data analysis section. The difference in AF incidence in the present study compared to prior ones is quite significant, but the authors diligently explain the reasons for this.

Points for revision:

MAJOR:

1) In the Results section, the statements of incidence comparison need to be substantiated with p-values. These are missing in both the text and table 2.

We have added these to the text and table 2 and also explain how they were calculated in the methods. Given the large effect sizes observed all p-values related to the effects of year, age and gender are <0.001. P-values for the effect of deprivation have now also been included in the main text.

2) The decline between 2015-18 seems to be more in the older age group 75-84 and 85+ than the younger ones. Did the authors note a significant change in the denominators of these groups between 2015-18? A differentiation should be made between drop in incidence due to a drop in gross number of new cases, versus increase in the denominator, or both.

We note that in the period 2015-2018 there is a substantial drop in the denominator, caused by a reduction in the number of contributing practices, however the drop in the numerator is larger than the drop in denominator in the older age groups. This is highlighted where we state: ‘changes in the number of practices using the Vision computer system in recent years could be contributing to the levelling off in incidence rates observed after 2015’.

3) In Table 1, the baseline characteristics for age and gender are provided. Given the extent of the available database, would it not be possible to include the frequencies of comorbidities associated with AF such as hypertension, congestive heart failure, stroke among others? This information could be useful in determining the reasons for the observed trends in AF incidence.

We have chosen not to add this as we feel this information is deviating from the aims of the paper. Knowing the prevalence of comorbidities in the AF cases *per se* would not contribute to understanding the observed trends. In order to do that we would have to examine how the prevalence of these comorbidities changes over time and how AF incidence changes over time in patients with these conditions; this would require a different study design and sample.

MINOR:

1) While the hypothesis of “boosting” incidence before 2015 can be entertained, the data provided is not confirming that this is the sole reason for the decline in AF incidence in more recent years. This is the first study to record a decline in AF incidence from 2015-18. While this is very interesting, the discussion should be expanded to include other potential reasons. In addition to health policy changes, the authors could comment on other factors such as improvement in technology of cardiac implanted device detected AF.

2) In the current era of wearable-device detected AF, this decline in AF incidence will likely not sustain in the future. The authors should mention a line on this in the implications for research section.

To address minor points 1 and 2, we have added the following text to the discussion:

Given the emergence of new technologies which allow self-diagnosis of atrial fibrillation, it is likely that the incidence of diagnosis of AF will rise again in the future. (REF1) Indeed, data from studies that

have used implantable loop recorders to detect AF suggest that there may be substantial proportions of people with short episodes of undiagnosed atrial fibrillation (REF2)

We have also amended the following sentence to:

It is important to continue to monitor trends in AF incidence to assess impact of new TECHNOLOGIES AND initiatives.

REF1: Khurshid S, Healey JS, McIntyre WF, Lubitz SA. Population based screening for atrial fibrillation. *Circulation Research* 2020; 127:143-154.

REF2: Diederichsen SZ, Haugan KJ, Brandes A, Lanng MB, Graff C, Krieger D et al. Natural history of subclinical atrial fibrillation detected by implanted loop recorders. *J Am Coll Cardiol* 2019; 74:2771-81.

Reviewer: 2

Reviewer Name: Eitaro Kodani

Institution and Country: Nippon Medical School Tama-Nagayama Hospital, Tokyo, Japan.

Please state any competing interests or state 'None declared': None

This manuscript by Mendonça et al. focused on the incidence atrial fibrillation (AF) in primary care. Authors investigated that the incidence of AF using a large cohort of 3.5 million patients over 40 years old registered in general practices in England and demonstrated its trends between 2004 and 2018 by sex and age classes. As authors mentioned, the prevalence and incidence of AF have been increasing along with increasing aged population, and recently become a social problem. Therefore, the concept of this study is valuable and results seem reasonable. Although this manuscript seems written well, authors may want to consider several issues as below.

Major comments;

1) How did you deal with the difference in observation periods among the year? For instance, was it for 14 years in the 2004's data? Whereas, how long was it 2018's data?

We calculated each year's incidence separately through the use of a lexis expansion. The lexis expansion splits each individual's follow up time into different year strata. To calculate the incidence for a specific year, we identify how much follow up time was contributed by all patients and how many AF cases were diagnosed for that specific year. AF cases no longer contribute time at risk to the denominator after their date of AF diagnosis. Patients with longer follow ups will contribute to multiple years until they deregister of their practice, die or have an AF diagnosis code recorded or their practice stops contributing.

2) In Figure 1, if authors wanted to compare the incidence of AF between the period of 2004-2010 and that of 2010-2018, show statistical difference.

We have now included in the results section and table 2 p-values for the effects of year, age and gender from unadjusted models. For the parameter year, specifically, the $p < 0.001$ tell us that year contributes significantly to explain the observed variation in incidence rates implying that there is a significant difference between some of the years. We prefer this approach rather than testing specific year categories which were not specified *a priori*. Specific comparisons between years can be inferred from the reported incidence rate confidence intervals.

3) In Appendix 6, show statistical difference.

We have added the p-value for the main effect of deprivation to the table caption in appendix 6.

Minor comments;

1) In abstract, abbreviations of NHS should be spelt out even in abstract. In line 2, although authors described that "The condition is easy to detect in primary care", I feel out of tune this sentence, because it seems difficult to detect paroxysmal AF in primary care. Authors may want to modify this sentence.

We have given the abbreviations in full, and changed the wording to: "The condition can be detected in primary care".

2) In main text, abbreviations of NHS, GRASP-AF, GOLD, and QOF should be spelt out at the first time of use.

Thank you, we have added these where they were missing except for “GOLD” which doesn’t appear to be an acronym.

3) In Box 1, abbreviations of QOF, NOAC, RCT, and NICE should be explained in footnote.

Thank you, these have been added.

4) In Appendix 4 and 5, please explain shortly what “Modelled rate” is in footnote.

We have added a short sentence: “Modelled rates take into account the effect of changes in the age-gender case mix over time.”

5) In Appendix 7, abbreviations of IMD should be spelt out.

Thank you, we have added this to the captions for both appendices 6 and 7. We have also added a sentence to the caption of appendix 7 to explain the use of IRR – “Relative differences between deprivation categories are shown as incidence rate ratios (IRR).” This is explained in the footnote in more detail.

VERSION 2 – REVIEW

REVIEWER	Eitaro Kodani Nippon Medical School Tama Nagayama Hospital, Tokyo, Japan
REVIEW RETURNED	06-Nov-2020
GENERAL COMMENTS	This revised manuscript by Mendonça et al. focused on the incidence atrial fibrillation (AF) in primary care. Authors revised the manuscript appropriately with statistical analyses according to the reviews’ comments. It appeared better.